# Improved Model-based Learning with Data Augmentation for Quantitative Susceptibility Mapping

**Juan Liu**                              JUAN.LIU@YALE.EDU

*Department of Radiology and Biomedical Engineering, Yale University, New Haven, CT, USA*

## Abstract

Quantitative susceptibility mapping (QSM) is a magnetic resonance imaging (MRI) technique that estimates magnetic susceptibility of tissue from MR phase measurements. Recently, several supervised deep learning (DL) techniques have demonstrated impressive performance in solving the challenging ill-posed field-to-source inverse QSM reconstruction problem. To address the lack of the inherent non-existent ground-truth QSM references, a model-based method was recently proposed using the well-established physical model. However, it fails to perform well at the regions with large susceptibility variations. Here, we proposed uQSM+ with data augmentation techniques to improve the model-based learning. The proposed method was evaluated on a multi-orientation QSM datasets and 2019 QSM reconstruction challenge datasets. Quantitative and qualitative evaluation showed that uQSM+ and zero-shot uQSM+ was capable of reconstructing high quality QSM. The code is available at https://github.com/juana313/uQSM-plus.

**Keywords:** Quantitative susceptibility mapping, Self-supervised learning.

## 1. Introduction

Quantitative susceptibility mapping (QSM) can estimate tissue magnetic susceptibility values from magnetic resonance imaging (MRI) Larmor frequency sensitive phase images (Wang and Liu, 2015). Biological tissue magnetism can provide useful diagnostic image contrast and be used to quantify biomarkers including iron, calcium, and gadolinium (Wang and Liu, 2015). To date, all QSM methods rely on a dipolar convolution that relates susceptibility sources to induced Larmor frequency offsets (Salomir et al., 2003; Marques and Bowtell, 2005), which is expressed in the k-space as bellow.

$$B(\boldsymbol{k}) = X(\boldsymbol{k}) \cdot D(\boldsymbol{k}); D(\boldsymbol{k}) = \frac{1}{3} - \frac{k_z^2}{k_x^2 + k_y^2 + k_z^2} \tag{1}$$

where $B(\boldsymbol{k})$ is the susceptibility induced magnetic perturbation along the main magnetic field direction $\boldsymbol{z}$, $X(\boldsymbol{k})$ is the susceptibility distribution $\chi$ in the k space, $D(\boldsymbol{k})$ is the dipole kernel. While the source-to-field forward relationship of this model is well-established and can be efficiently computed using Fast-Fourier-Transform (FFT), the k-space singularity in the dipole kernel results in an ill-conditioned relationship in the field-to-source inversion.

Calculation of susceptibility through multiple orientation sampling (COSMOS) (Liu et al., 2009) remains the empirical gold-standard of QSM, as the additional field data sufficiently improves the conditioning of the inversion algorithm. Since it is time-consuming and clinically infeasible to acquire multi-orientation data, single-orientation QSM is preferred which is computed by either thresholding of the convolution operator (Shmueli et al.,

2009; Wharton et al., 2010; Haacke et al., 2010) or use of more sophisticated regularization methods (De Rochefort et al., 2008; de Rochefort et al., 2010; Liu et al., 2011; Bilgic et al., 2014). In single-orientation QSM, inaccurate field-to-source inversion often causes large susceptibility quantification errors that appear as streaking artifacts, especially in massive hemorrhagic regions.

Recently, several deep learning (DL) approaches have been proposed to solve for the QSM dipole inversion, such as QSMnet (Yoon et al., 2018), DeepQSM (Bollmann et al., 2019), FINE (Zhang et al., 2019), QSMGAN (Chen et al., 2019), QSMnet+ (Jung et al., 2020), and xQSM (Gao et al., 2021). These DL techniques have exhibited impressive results and demonstrated the superiority of DL to address the challenging QSM reconstruction problem. These methods are supervised and data-driven which require QSM labels for network training. Unfortunately, QSM has the inherent non-existent 'ground-truth'. Therefore, these methods use either COSMOS data or synthetic data for network training. However, acquiring a large number of COSMOS data is not only expensive but also time consuming. Moreover, COSMOS does not ensure the model consistency and contains errors from image registration procedures. Though synthetic data provides a reliable and cost-effective way for training, the generalization capability needs to be addressed when applying on real data. To overcome these limitations, Liu et al.(Liu and Koch, 2020) proposed a model-based learning method, denoted as uQSM, without the need of QSM labels for network training.

Though the model-based learning method showed promising, uQSM fails to perform well at the regions with large susceptibility variations. To address this issue, we propose a data augmentation technique to regularize the model-based training. The method, denoted as uQSM+, uses random pseudo high susceptibility sources induced field perturbation to superimpose on the original local field, to improve the robustness of model-based learning. We conduct extensive experiments on a multi-orientation datasets and 2019 QSM reconstruction challenge. The results show that uQSM+ and **'zero-shot'** uQSM+ (zs-uQSM+) consistently leads to significant improvements.

## 2. Method

Due to that the field-to-source inversion is constrained by the physical model in Eq.1, the conventional data augmentation techniques such as random rotation, shearing, and color change etc is not applicable. Here we proposed a novel way to bring implicit regularization to increase the robustness of the unsupervised training. Assuming that the network is trained to perform well to derive the susceptibility map $\chi$ from the local field $f$, it should generalize well to derive the susceptibility map $\chi + \chi_b$ from the perturbed local field $f + d * \chi_b$, where $d$ is the dipole kernel, $*$ is the convolution operator, $\chi_b$ is perturbed susceptibility map, $f_b = d * \chi_b$ is the induced perturbed field. Based on that uQSM performs poor at regions with large susceptibility variations, we intentionally generate the perturbed field using high susceptibility sources to mimic the bleeding (with large positive susceptibility values) and calcification (with large negative susceptibility values).

Fig.1 shows the network architecture of uQSM+. uQSM+ adopts a 3D U-Net like architecture (Ronneberger et al., 2015). Different from QSMnet and DeepQSM, uQSM+ applies only one convolutional layers at each level of encoding/decoding scales, to make the network smaller to be able to accept larger patch size (more contextual information)

during network training. In addition, the batch normalization layers are discarded here, since the batch size in training is small, which could influence the effectiveness of batch normalization. We found that uQSM+ without batch normalization layers can be trained faster at the same time without sacrificing the performance.

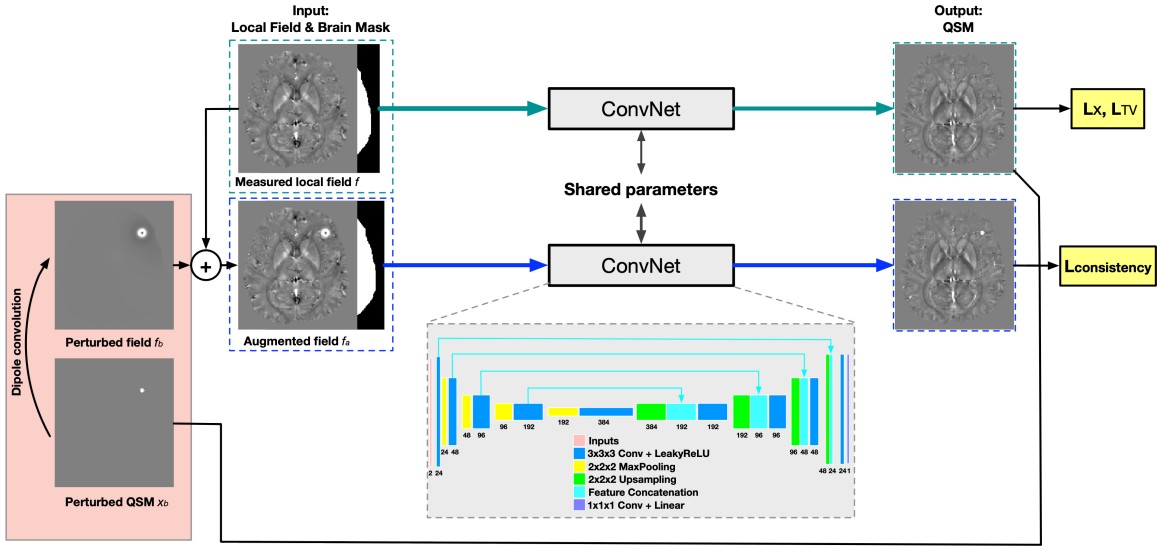

Figure 1: Neural network architecture of uQSM+.

During network training, the network takes the local field measurement $f$ and the brain mask $m$, and at the same time the augmented local field measurement $f_a = f + d * \chi_b$ and the brain mask $m$, to to infer $\chi$ and $\chi_a$ respectively. The data relationship between $\chi$ and $\chi_a$ is used as a regularization term in the loss function.

The loss function incorporates the data fidelity loss $L_\chi$. From uQSM, the fidelity loss using nonlinear dipole inversion performs better than the fidelity loss using linear dipole inversion and weighted linear dipole inversion. Therefore, here we also use the nonlinear dipole inversion in the loss function.

$$L_\chi = \left\| W m (e^{jd*\chi} - e^{jf}) \right\|_2 \tag{2}$$

where $W$ serves as a data-weighting factor which can be the magnitude image or noise weight matrix.

$$L_{TV} = \|G_x(\chi)\|_1 + \|G_y(\chi)\|_1 + \|G_z(\chi)\|_1 \tag{3}$$

In addition, a total variation (TV) loss $L_{TV}$ on the output $\chi$ is to preserve important details such as edges whilst removing unwanted noise in the reconstructed susceptibility maps. In $L_{TV}$, $G_x$, $G_y$, $G_z$ are gradient operators in x, y, z directions.

For $\chi$ and $\chi_a$, the susceptibility difference inside the pseudo bleeding/calcification regions should be the susceptibility values of pseudo bleedings/calcification, and the susceptibility

difference outside of the pseudo bleeding regions should be zeros so that no artifacts appears. Therefore, two data consistency losses are defined,

$$L_{consist\_i\_ROI} = \|m_{ROI}((\chi_a - \chi) - \chi_b))\|_2 \tag{4}$$

$$L_{consist\_x\_ROI} = \|m_{XROI}(\chi_a - \chi)\|_2 \tag{5}$$

where $m_{ROI}$ is the regions of pseudo bleeding and calcifications, $m_{XROI}$ is the regions outside of bleeding and calcifications.

$$L_{Total} = L_\chi + \lambda_1 L_{TV} + \lambda_2(t)L_{consist\_i\_ROI} + \lambda_3(t)L_{consist\_x\_ROI} \tag{6}$$

The loss function is the weighted sum of the above four loss terms. $\lambda_2(t)$ and $\lambda_3(t)$ are changed in ramp-up (i.e., a gradually increasing weight) then ramp-down (i.e., a gradually decreasing weight) strategy. At the beginning of network training, the network starts to learn for dipole inversion, therefore $\lambda_2(t)$ and $\lambda_3(t)$ are set low. Then $\lambda_2(t)$ and $\lambda_3(t)$ gradually increase to impose larger regularization. At the last few epochs, $\lambda_2(t)$ and $\lambda_3(t)$ gradually decrease so that the network learns to better meet the physical model. At the network prediction stage, the trained model only takes $f$ and mask $m$ to infer $\chi$.

Further, we investigate **"zero-shot"** uQSM+ (**zs-uQSM+**) to perform single data QSM reconstruction, which does not rely on training data for prior training.

## 3. Experiments

**Multi-orientation Datasets** 9 QSM datasets were acquired using 5 head orientations and a 3D single-echo GRE scan with isotropic voxel size 1.0x1.0x1.0 mm$^3$ on 3T MRI scanners. QSM data processing was implemented as following, offline GRAPPA (Griswold et al., 2002) reconstruction to get magnitude and phase images from saved k-space data, coil combination using sensitivities estimated with ESPIRiT (Uecker et al., 2014), BET (FSL, FMRIB, Oxford, UK) (Smith, 2002) for brain extraction, Laplacian method (Li et al., 2011) for phase unwrapping, and RESHARP (Wu et al., 2012) with spherical mean radius 4mm for background field removal. COSMOS results were calculated using the 5 head orientation data which were registered by FLIRT (FSL, FMRIB, Oxford, UK) (Jenkinson et al., 2002; Jenkinson and Smith, 2001).

For uQSM+ training, leave-one-out cross validation was used. For each dataset, total 40 scans (8*5) from other 8 datasets were used for training. uQSM+ was trained on patch-based with patch size 96x96x96. The RESHARP local field and brain mask patches with patch size 96x96x96 were randomly cropped during training. The magnitude images were used as the weighting factor $W$. 100 perturbed susceptibility maps with random ellipsoid shapes (transverse, equatorial and polar radii 1-5mm), random rotation in three axes (0-360°), random susceptibility value ($N(u = \pm1.5ppm, \sigma = 0.1ppm)$), random location were generated. Adam optimizer (Kingma and Ba, 2014) was used for the model training. The initial learning rate was set as 0.0002, with exponentially decay at every 200 steps. The network was trained with 15 epochs, 200 steps for each epoch, batch size 2. The model was trained and evaluated using Tensorflow 2.2. For zs-uQSM+ implementation, we also used

96x96x96 patch-based with random cropped patches from the single dataset itself, with total 750 iterations.

In addition, QSM estimates were generated using the TKD (Shmueli et al., 2009), MEDI (Liu et al., 2012), deep image prior (DIP), and uQSM. The results of TKD, MEDI, uQSM, uQSM+, zs-uQSM+ were compared with respect to the COSMOS maps using quantitative metrics, peak signal-to-noise ratio (PSNR), normalized root mean squared error (NRMSE), high frequency error norm (HFEN), and structure similarity (SSIM) index.

**2019 QSM Reconstruction Challenge** On the 2019 QSM reconstruction challenge stage 2 (http://qsm.snu.ac.kr/?pageid=30), four datasets with two contrast levels and two noise levels (denoted by Sim1Snr1, Sim1Snr2, Sim2Snr1, and Sim2Snr2, with "Sim" representing contrast level and "Snr" standing for noise level) were generated using MR simulation(Marques et al., 2021; Bilgic et al., 2020). The metrics for evaluation in the challenge are NRMSE, dNRMSE (detrended NRMSE), dNRMSE Tissue, dNRMSE DeepGM, NRMSE Blood, DLS (Deviation From Linear Slope), Calcification Streak, and Deviation From Calcification Moment (Calcification Error).

Since each dataset has different contrast level and noise level, zs-uQSM+ was used to get the QSM result of each dataset. At first stage, the image patches with patch size 96x96x96 were randomly cropped during training. 100 perturbed susceptibility maps with fake bleedings ($N(u = 2.0ppm, \sigma = 0.1ppm)$) and calcifications ($N(u = -2.0ppm, \sigma = 0.1ppm)$) and their induced local field were generated. The noise weight matrix was scaled to $[0, 1]$ and used as $W$ in the loss function. After 2000 iterations, the full field map with image size 160x160x160 was inputted to network for further fine-tuning with another 2000 iterations using the loss $L_{Total} = L_\chi + \lambda_1 L_{TV}$. For dataset with different noise levels, $\lambda_1$ was set as 0.004 and 0.002 for noise levels Snr1 and Snr2, respectively.

## 4. Experimental Results

**Multi-orientation Datasets** Table.1 summarized quantitative metrics on 9 multi-orientation datasets. Compared to TKD, MEDI, and uQSM, uQSM+ results achieved the best metric scores in PSNR, RMSE, and HFEN, and second best in SSIM.

Table 1: Means and standard deviations of quantitative performance metrics of 9 reconstructed QSM images with COSMOS as a reference on 9 multi-orientation datasets.

|  | PSNR (dB) | NRMSE (%) | HFEN (%) | SSIM (0-1) |
|---|---|---|---|---|
| **TKD** | $43.4 \pm 0.5$ | $91.4 \pm 6.7$ | $72.9 \pm 6.6$ | $0.831 \pm 0.016$ |
| **MEDI** | $41.5 \pm 0.6$ | $113.8 \pm 7.6$ | $100.4 \pm 9.1$ | **$0.902\pm0.016$** |
| **DIP** | $44.0 \pm 0.8$ | $85.5 \pm 6.7$ | $65.7 \pm 4.5$ | $0.859 \pm 0.020$ |
| **uQSM** | $45.6\pm0.4$ | $71.4\pm5.0$ | $62.8\pm5.0$ | $0.890 \pm 0.015$ |
| **uQSM+** | **$46.1\pm0.5$** | **$67.2\pm3.9$** | **$59.6\pm3.4$** | $0.892 \pm 0.012$ |
| **zs-uQSM+** | $46.0\pm0.5$ | $67.5\pm3.8$ | $60.4\pm4.5$ | $0.887 \pm 0.012$ |

Fig.2 compared QSM images from a representative dataset. Streaking artifacts were observed in TKD, MEDI, and DIP results (a-c, iii, black solid arrows). uQSM showed

shadow artifacts around the large vessels (d, iii, black arrow). Compared with uQSM, uQSM+ and zs-uQSM+ results displayed better image quality and suppressed shadows.

Figure 2: Comparison of QSM of a multi-orientation data. TKD (a), MEDI (b), and DIP (c) maps showed oversmoothing and/or streaking artifacts. The uQSM (d) maps well preserve image details but show black shading artifacts. uQSM+ (e) and zs-uQSM+ (f) achieved impressive quality and suppressed shadows.

**2019 QSM reconstruction challenge** Table.2 showed the quantitative metrics scores of zs-uQSM+ on 2019 QSM reconstruction challenge stage 2. When compared with all 17 submitted DL-based methods on SNR1 (Appendix Table.4 and Table.5), zs-uQSM+ achieved the second best in NRMSE, dNRMSE tissue, dNRMSE blood, dNRMSE DGM, CalcStreak, CalcificationError, fourth best in DLS. On SNR2, zs-uQSM+ achieved the second best in NRMSE, dNRMSE tissue, dNRMSE blood, dNRMSE DGM, CalcStreak, CalcificationError, and DLS.

Table 2: Quantitative metrics of zs-uQSM+ on 2019 QSM challenge datasets.

|  | **NRMSE** | **dNRMSE** | **dNRMSE Tissue** | **dNRMSE Blood** |
|---|---|---|---|---|
| **SNR1** | 40.349 | 42.723 | 47.586 | 67.143 |
| **SNR2** | 38.364 | 40.058 | 44.861 | 62.501 |

|  | **dNRMSE DGM** | **DLS** | **CalcStreak** | **CalcificationError** |
|---|---|---|---|---|
| **SNR1** | 26.256 | 0.0656 | 0.0267 | 7.993 |
| **SNR2** | 24.606 | 0.0532 | 0.0285 | 10.981 |

Fig.3 displayed the QSM results of QSM challenge 2019 datasets stage2 Sim2Snr2. Streaking artifacts were clearly visible in submitted TKD, DeepQSM, xGAN, CAD QSM-net results. MEDI and FINE achieved impressive quality and high scores of quantitative metrics. The proposed zs-uQSM+ displayed high quality and invisible artifacts, but showed enlarged calcification regions when compared with the ground truth.

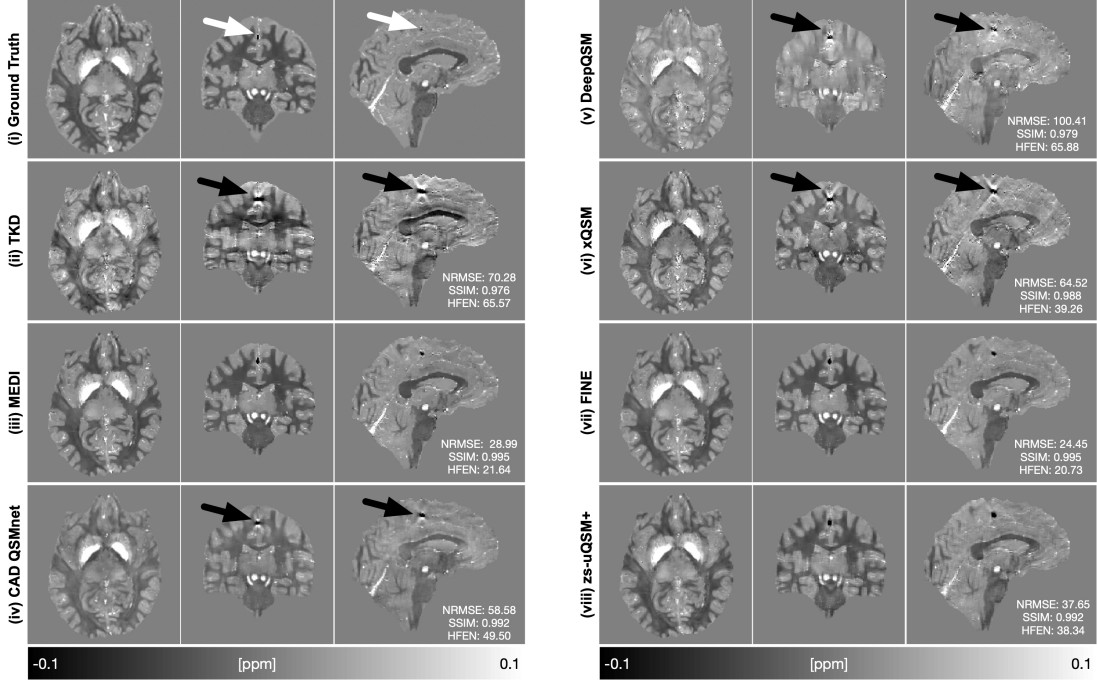

Figure 3: QSM results on 2019 QSM reconstruction challenge stage2 Sim2Snr2. Streaking artifacts show up around the calcification region in the submitted results of TKD, QSMnet, DeepQSM, xQSM, indicated by black arrows.

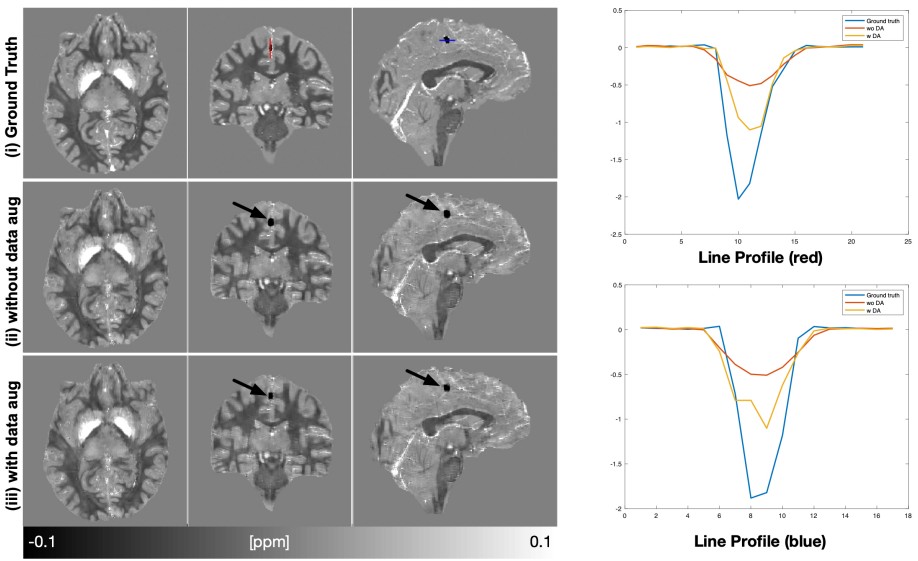

Figure 4: Illustration of the effects of the data augmentation on 2019 QSM reconstruction challenge stage2 Sim2Snr2 dataset.

Fig.4 showed that the results with and without data augmentation on Sim2Snr2. Without the proposed data augmentation, the result showed invisible image artifacts while enlarged calcification regions (ii). From the line profile, zs-uQSM+ better estimated the susceptibility values at calcification region (yellow) than that without data augmentation (red), but still failed to accurately estimate the susceptibility values.

## 5. Discussion

In this work, an improved model-based learning for QSM was proposed which utilize the local field perturbation to introduce regularization in the unsupervised learning. From quantitative evaluation and visual assessment on multi-orientation QSM datasets and 2019 QSM reconstruction challenge datasets, uQSM+ and zs-uQSM+ achieved impressive performance. On 2019 QSM reconstruction challenge, uQSM+ outperformed many supervised learning methods. We think that many submitted supervised DL-based methods neglected the problem of domain shift induced performance drop. The reason that FINE achieved impressive performance might be that it was trained with the data itself MEDI result, in which the well-tuned MEDI result performed pretty well and using the data itself MEDI result for training does not have the domain shift problem.

Compared with uQSM, uQSM+ utilizes local field perturbation to better regularize the unsupervised learning for dipole inversion. The local field perturbation explicitly imposes regularization and improve the robustness and generalization of network. Compare with other proposed supervised DL methods such as QSMnet, DeepQSM, QSMGAN, etc, uQSM+ and zs-uQSM+ were trained in an unsupervised way without the need of QSM labels. The unsupervised learning has the advantage of easy preparing training data.

Although the proposed uQSM+ improved the model-based learning for QSM, there are still exist some limitations. First, uQSM+ is still affected by the processing steps ahead QSM inversion, such as field estimation, phase unwrapping, and background field removal methods. It is necessary to investigate these effects on susceptibility quantification. Second, at regions with extreme large susceptibility variation such as 2019 QSM reconstruction challenging datasets, uQSM+ has the difficulty to accurately quantify the susceptibility values at the calcification region. The possible reason is that the network has difficulty to learn the high frequency components of a signal due to spectral bias. Further study will investigate and address this problem. Third, the patch-based network training cannot fully guarantee the data fidelity.

## 6. Conclusion

In conclusion, we proposed an improved model-based learning for QSM which use the perturbation of the local field to bring regularization and improve the robustness of solving the ill-posed field-to-source inverse problem. The proposed method can give promising results in terms of artifacts and image quality. We believe that uQSM+ and zs-uQSM+ could serve a good baseline for DL-based QSM techniques in the future.

## Acknowledgments

We thank Professor Jongho Lee for sharing the multi-orientation QSM datasets.

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

## Appendix A. Multi-orientation datasets

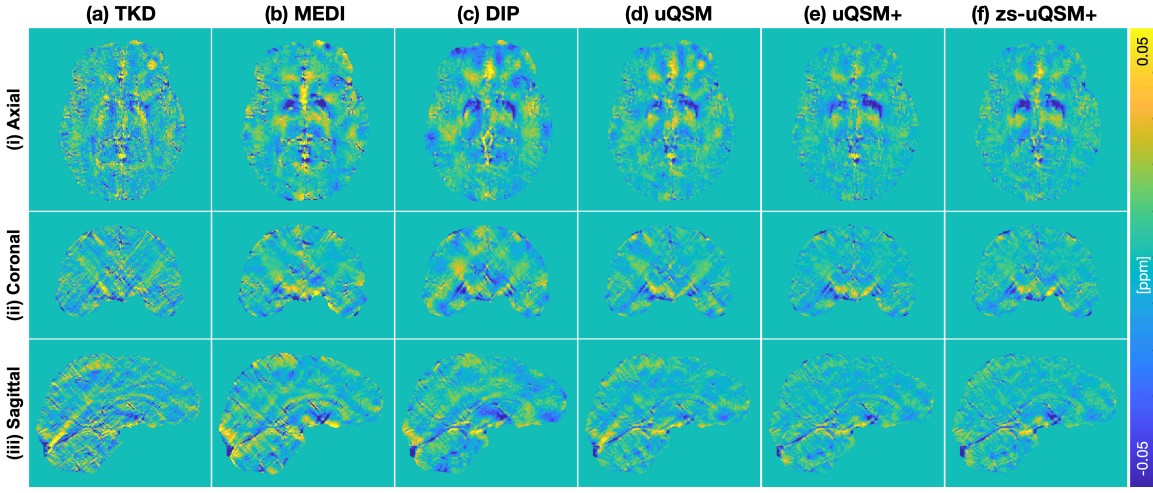

Figure 5: Residual error maps of Fig.2 with COSMOS as a reference. From the coronal and sagittal planes, uQSM+ and zs-uQSM+ have less streaking residuals.

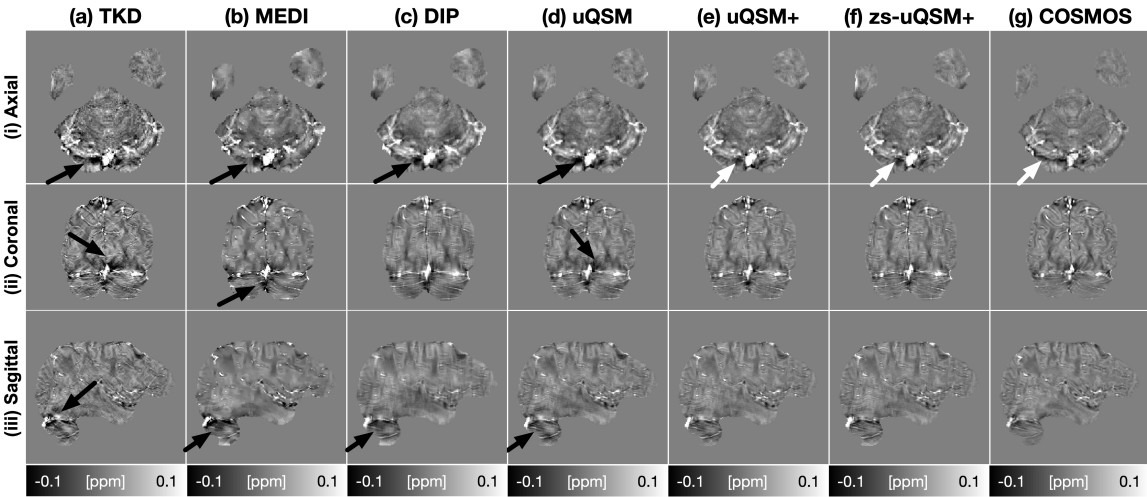

Figure 6: Comparison of QSM of a multi-orientation data. Compared with TKD, MEDI, DIP, and uQSM, uQSM+ (e) and zs-uQSM+ (f) greatly suppressed the shadow artifacts and achieved impressive image quality.

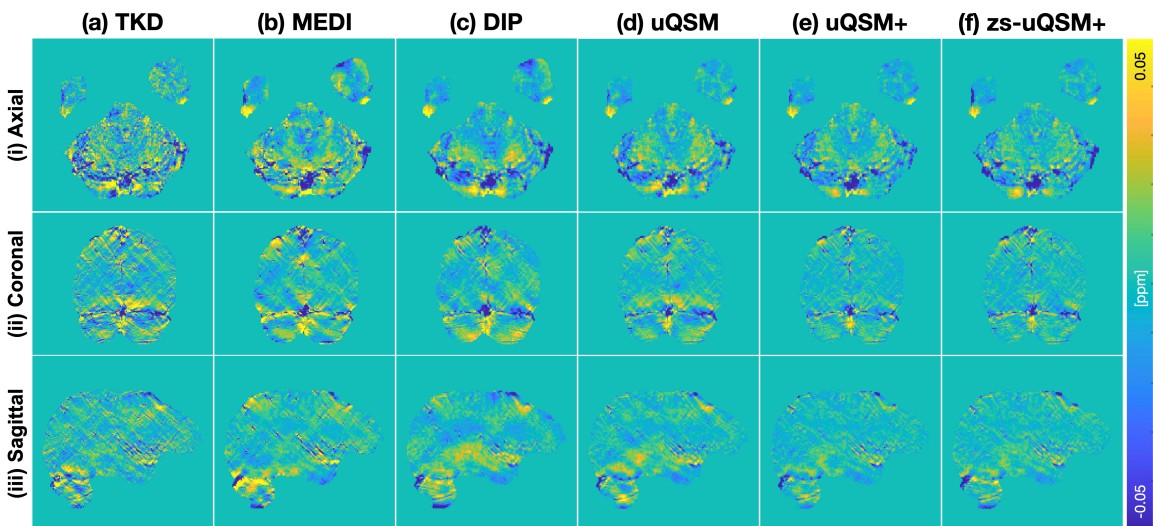

Figure 7: Residual error maps of Fig.6 with COSMOS as a reference.

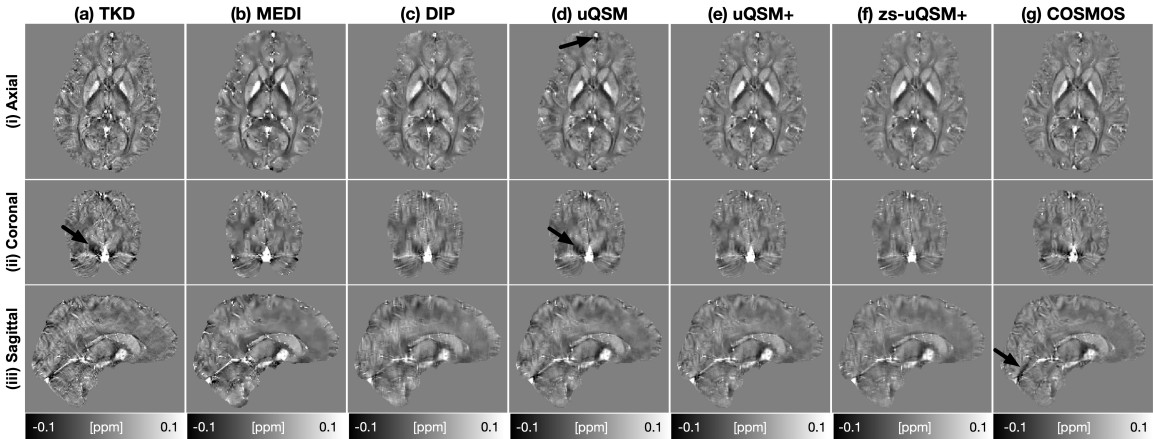

Figure 8: Comparison of QSM of a multi-orientation data. Compared with TKD, MEDI, DIP, and uQSM, uQSM+ (e) and zs-uQSM+ (f) greatly suppressed the shadow artifacts and achieved impressive image quality.

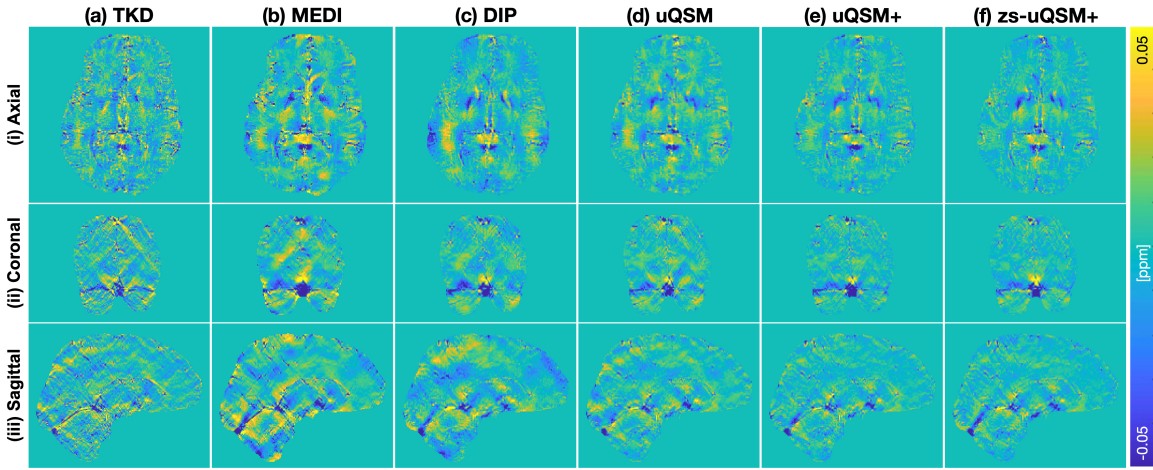

Figure 9: Residual error maps of Fig.8 with COSMOS as a reference.

## Appendix B. Comparison of other data augmentation

Two data augmentation techniques are compared with the proposed data augmentation technique.

(1) applying random flipping in x, y directions, assuming the main magnetic field along the z direction.

(2) adding Gaussian noise in the local field input. Let $f_n = f + n$, where $f_n$ is the local field with adding Gaussian noise, $f$ is the calculated local field without adding Gaussian

noise, $n$ is the the added Gaussian noise. The loss function is weighted sum of $L_\chi = \left\|Wm(e^{jd*\chi} - e^{jf})\right\|_2$ and $L_{TV}$.

Table 3: Means and standard deviations of quantitative performance metrics of 9 reconstructed QSM images with COSMOS as a reference on 9 multi-orientation datasets.

|                   | PSNR (dB)   | NRMSE (%)   | HFEN (%)    | SSIM (0-1)          |
|-------------------|-------------|-------------|-------------|---------------------|
| **uQSM**          | 45.6±0.4    | 71.4±5.0    | 62.8±5.0    | 0.890 ± 0.015       |
| **uQSM w flipping** | 45.8 ± 0.7 | 69.8 ± 6.4  | 62.5 ± 6.7  | 0.890 ± 0.013       |
| **uQSM w noise**  | 45.6±0.7    | 71.3±7.8    | 63.7±7.0    | 0.892 ± 0.016       |
| **uQSM+**         | **46.1±0.5** | **67.2±3.9** | **59.6±3.4** | **0.892±0.012**   |

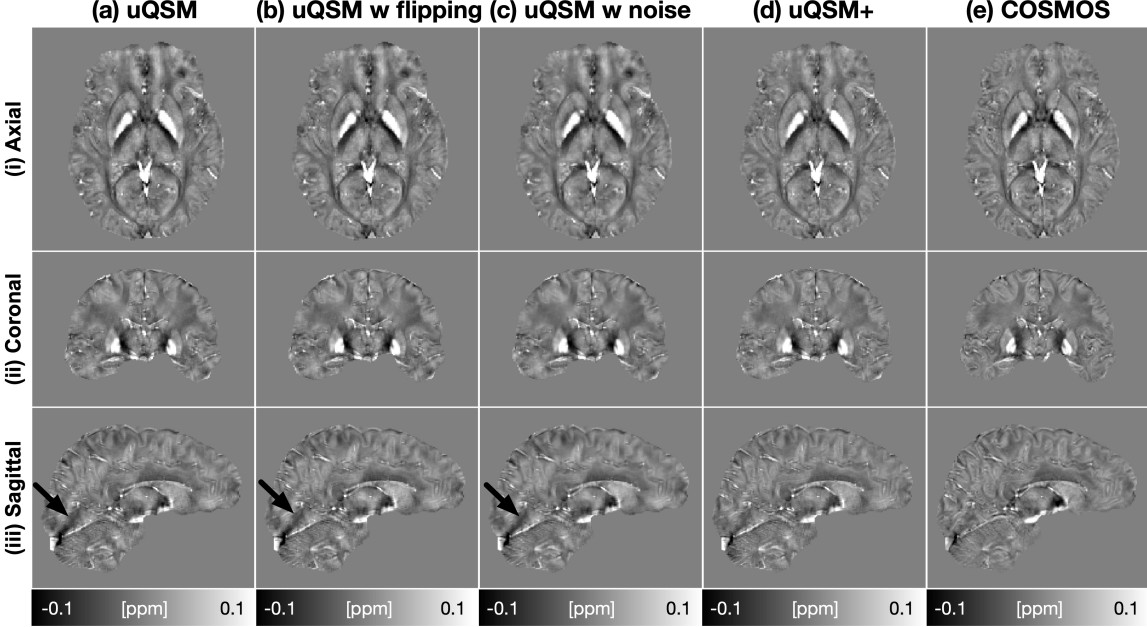

Figure 10: Comparison of three data augmentation methods. In (b) and (c), uQSM with flipping and adding noise reconstruct QSM with artifacts (black arrows). The proposed method (d) better suppressed the shadow artifacts.

Table.3 showed that the proposed data augmentation achieved the best quantitative metric scores among two other data augmentation techiniques. Fig.10 showed that data augmentation using flipping and adding Gaussian noise cannot effectively improve the QSM result. The proposed method is much more effective for suppressing artifacts.

## Appendix C. Comparison of different geometric shapes of perturbed sources

In the paper, we used ellipsoid shape to generate the perturbed susceptibility sources. To know whether the geometric shapes could effects the performance of uQSM+, we compared four geometric shapes - (1) cuboid, (2) sphere, (3) cylinder and (4) ellipsoid. The multi-orientation datasets were used for comparison. For PSNR, there is no significant difference (ellipsoid vs cuboid $p = 0.27$, ellipsoid vs cylinder $p = 0.31$, ellipsoid vs sphere $p = 0.22$). For NRMSE, there is no significant difference (ellipsoid vs cuboid $p = 0.40$, ellipsoid vs cylinder $p = 0.46$, ellipsoid vs sphere $p = 0.36$). For HFEN, there is no significant difference (ellipsoid vs cuboid $p = 0.39$, ellipsoid vs cylinder $p = 0.47$, ellipsoid vs sphere $p = 0.31$). For SSIM, there is no significant difference (ellipsoid vs cuboid $p = 0.41$, ellipsoid vs cylinder $p = 0.44$, ellipsoid vs sphere $p = 0.97$). In addition, based on visual assessment, there is no difference among them.

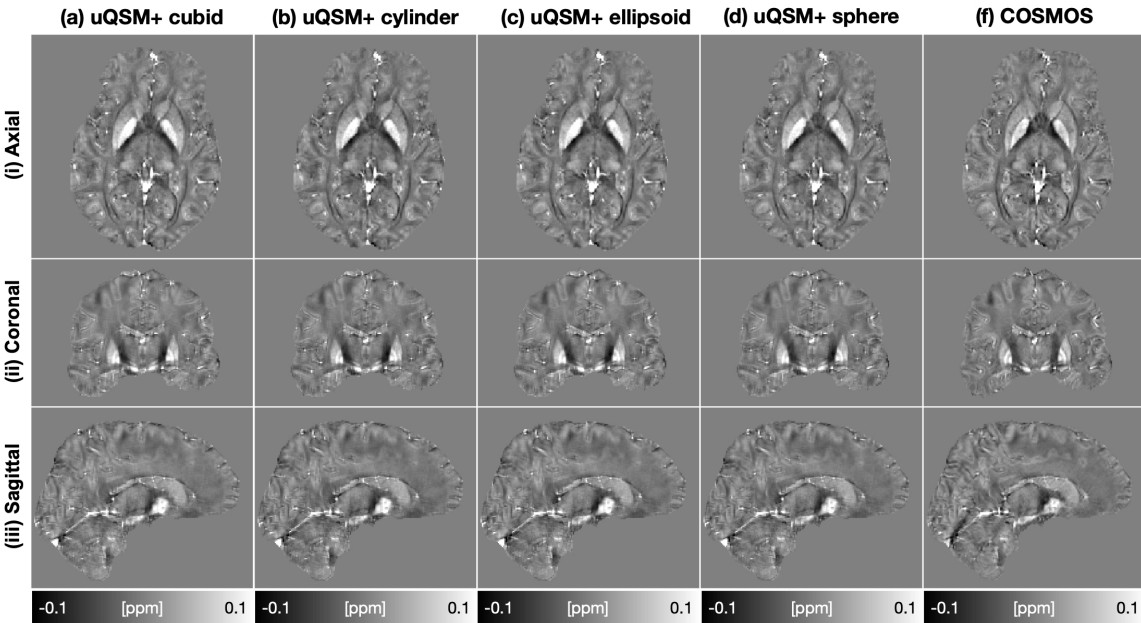

Figure 11: Comparison of different geometric shapes of perturbed sources. Based on visual assessment, uQSM+ using these four geometric shapes of perturbed sources reconstruct comparable QSM.

## Appendix D.  zs-QSM+ on 2019 QSM reconstruction challenge stage 2

Figure 12: zs-uQSM+ results on 2019 QSM reconstruction challenge stage2.

## Appendix E.  Effects of weighting factor on 2019 QSM Challenge data stage2

The 2019 QSM reconstruction challenge datasets were generated using 4 echo time MR simulation. Here we compared the QSM reconstruction results when choosing the different weighting factor used in $L_\chi = \left\| Wm(e^{jd*\chi} - e^{jf}) \right\|_2$. (1) 1st echo magnitude image, (2) 2nd echo magnitude image, (3) 3rd echo magnitude image, (4) 4th echo magnitude image, (5) noise weighting matrix which was obtained when doing nonlinear field map fitting from the multi-echo magnitude and phase images using MEDI toolbox.

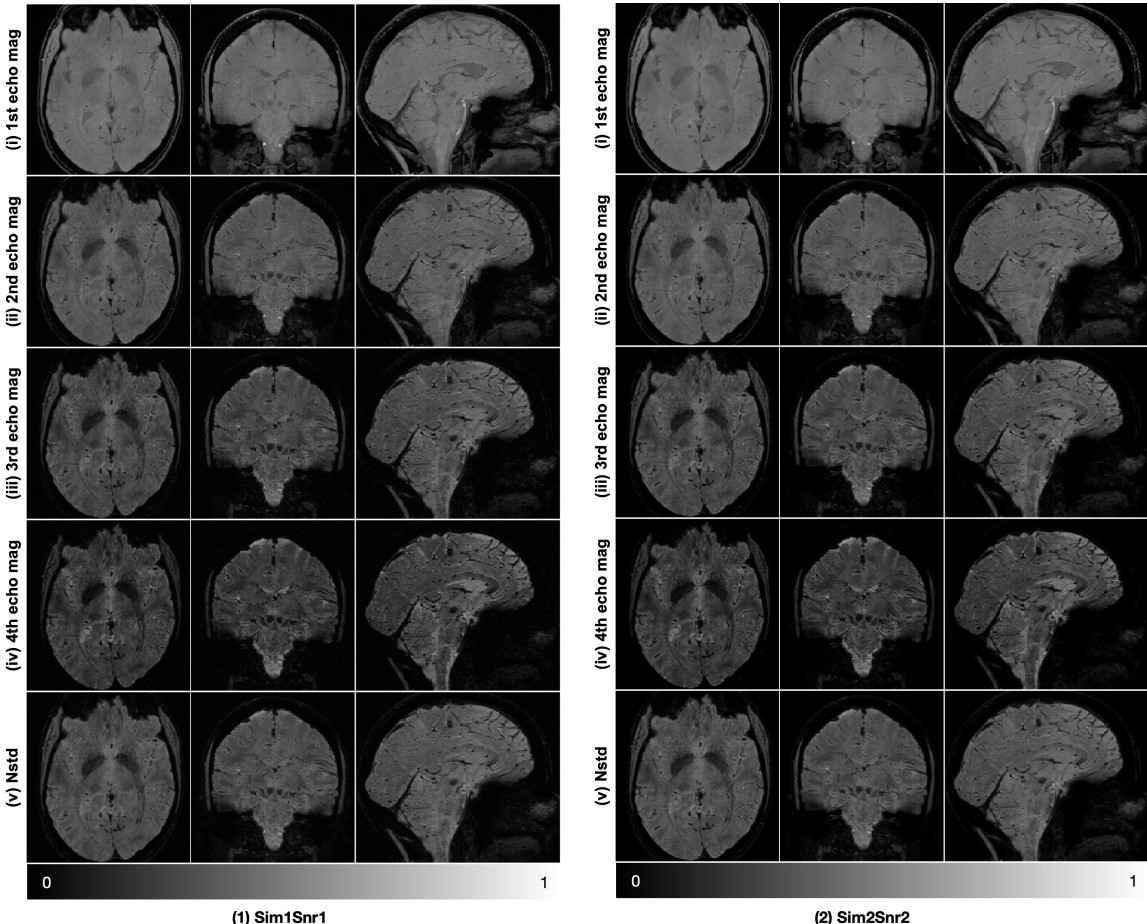

Figure 13: Illustration of magnitude images and noise weighting image of Sim1SNR1 and Sim2SNR2, with scaling to [0, 1]. The Sim1SNR1 has higher noise level than Sim2SNR2. The image contrast of magnitude images increases from 1st echo to 4th echo. The noise weighting images also similar image contrast with 3rd echo magnitude image, with higher noise level at calcification region, vessels, and globus pallidus, which show darker in the image.

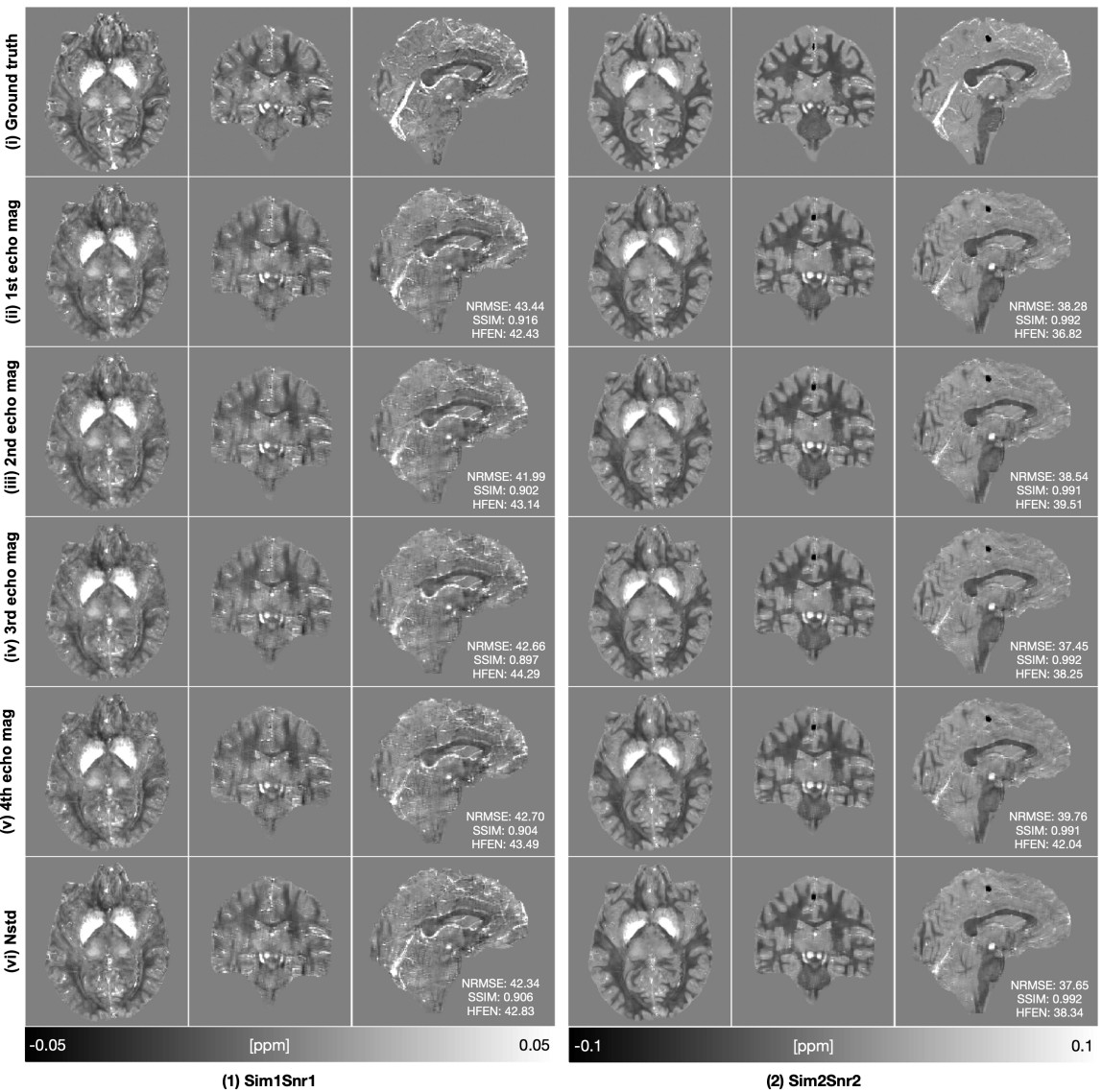

Figure 14: Illustration of QSM of Sim1SNR1 and Sim2SNR2 using different weighting factor in zs-QSM+. Based on visual assessment, zs-QSM+ using different weighting factor generates comparable QSM.

# Appendix F. Deep learning methods on 2019 QSM Reconstruction Challenge stage 2

Table 4: Quantitative metrics of submitted deep learning methods on 2019 QSM challenge stage2 SNR1.

| Identifier | NRMSE | dNRMSE | dNRMSE Tissue | dNRMSE Blood |
|---|---|---|---|---|
| 9uErOsxGtr | 60.95 | 76.45 | 95.40 | 148.64 |
| BZA1ZHG0K7 | 61.78 | 78.87 | 100.53 | 152.92 |
| cbi99GQHhl | 94.23 | 108.60 | 147.31 | 163.90 |
| FTAaNyeZiW | 58.28 | 71.78 | 90.37 | 132.41 |
| Jg4olcpuFP | 50.29 | 55.97 | 66.46 | 94.16 |
| Mx2TDToh2i | 86.65 | 113.15 | 142.27 | 181.01 |
| NF3tlGX8ED | 61.57 | 77.64 | 90.96 | 127.33 |
| nxEaOyG7or | 63.18 | 82.01 | 107.65 | 153.84 |
| OGggRA8FeM | 91.16 | 118.42 | 153.38 | 197.20 |
| rjEHqZ7N2X | 61.76 | 77.11 | 105.18 | 113.48 |
| SvA010zZI6 | 60.41 | 75.21 | 95.66 | 148.06 |
| u1qUWqXjWY | 86.87 | 102.86 | 132.05 | 156.48 |
| urVBXFAeow | 96.40 | 153.26 | 251.58 | 187.97 |
| VvuzvIauS6 | 64.41 | 83.81 | 107.77 | 161.47 |
| waXOIrgZog | 68.30 | 92.23 | 118.39 | 170.72 |
| BnhV1weeRf | 30.41 | 31.85 | 37.32 | 50.53 |
| YTQ7SppYwF | 66.97 | 86.04 | 106.78 | 163.44 |

| Identifier | dNRMSE DGM | DLS | CalcStreak | CalcificationError |
|---|---|---|---|---|
| 9uErOsxGtr | 43.27 | 0.0469 | 0.1130 | 40.17 |
| BZA1ZHG0K7 | 38.90 | 0.0179 | 0.1624 | 40.17 |
| cbi99GQHhl | 47.19 | 0.0749 | 0.2113 | 27.68 |
| FTAaNyeZiW | 32.93 | 0.1595 | 0.0935 | 36.47 |
| Jg4olcpuFP | 33.37 | 0.0814 | 0.1148 | 40.12 |
| Mx2TDToh2i | 59.07 | 0.1021 | 0.1133 | 36.97 |
| NF3tlGX8ED | 77.04 | 0.3458 | 0.1036 | 44.07 |
| nxEaOyG7or | 39.42 | 0.0630 | 0.1044 | 45.52 |
| OGggRA8FeM | 70.96 | 0.3302 | 0.1344 | 45.81 |
| rjEHqZ7N2X | 33.61 | 0.1294 | 0.0803 | 37.62 |
| SvA010zZI6 | 41.63 | 0.1561 | 0.1314 | 40.35 |
| u1qUWqXjWY | 54.12 | 0.0653 | 0.1089 | 26.52 |
| urVBXFAeow | 65.58 | 0.2580 | 0.1268 | 43.07 |
| VvuzvIauS6 | 45.48 | 0.1215 | 0.1061 | 45.97 |
| waXOIrgZog | 46.61 | 0.0316 | 0.1673 | 45.73 |
| BnhV1weeRf | 19.86 | 0.0212 | 0.0147 | 4.47 |
| YTQ7SppYwF | 44.20 | 0.1679 | 0.0667 | 48.81 |

Table 5: Quantitative metrics of submitted deep learning methods on 2019 QSM challenge stage2 SNR2.

| Identifier | NRMSE | dNRMSE | dNRMSE Tissue | dNRMSE Blood |
|---|---|---|---|---|
| 9uErOsxGtr | 63.25 | 78.21 | 99.43 | 151.20 |
| BZA1ZHG0K7 | 63.15 | 80.40 | 105.18 | 155.03 |
| cbi99GQHhl | 105.57 | 91.72 | 106.31 | 153.59 |
| FTAaNyeZiW | 57.17. | 69.70 | 86.71 | 127.77 |
| Jg4olcpuFP | 46.08 | 50.51 | 58.95 | 87.42 |
| Mx2TDToh2i | 73.07 | 92.63 | 113.48 | 157.31 |
| NF3tlGX8ED | 61.17 | 77.02 | 90.34 | 126.05 |
| nxEaOyG7or | 64.64 | 83.49 | 116.58 | 152.22 |
| OGggRA8FeM | 80.92 | 106.69 | 141.50 | 180.90 |
| rjEHqZ7N2X | 61.15 | 76.00 | 102.58 | 112.51 |
| SvA010zZI6 | 61.72 | 77.18 | 101.79 | 149.42 |
| u1qUWqXjWY | 70.07 | 82.47 | 104.01 | 133.06 |
| urVBXFAeow | 85.84 | 132.16 | 204.96 | 165.55 |
| VvuzvIauS6 | 65.70 | 86.35 | 116.70 | 159.30 |
| waXOIrgZog | 68.51 | 93.14 | 121.98 | 167.87 |
| BnhV1weeRf | 28.260 | 29.37 | 33.84 | 47.86 |
| YTQ7SppYwF | 64.74 | 81.72 | 100.77 | 155.60 |

| Identifier | dNRMSE DGM | DLS | CalcStreak | CalcificationError |
|---|---|---|---|---|
| 9uErOsxGtr | 41.54 | 0.0289 | 0.0959 | 39.39 |
| BZA1ZHG0K7 | 38.70 | 0.0368 | 0.1345 | 41.84 |
| cbi99GQHhl | 45.25 | 0.0340 | 0.2212 | 26.04 |
| FTAaNyeZiW | 32.71 | 0.1573 | 0.1084 | 23.38 |
| Jg4olcpuFP | 31.36 | 0.0761 | 0.1141 | 39.87 |
| Mx2TDToh2i | 51.11 | 0.1010 | 0.1244 | 33.83 |
| NF3tlGX8ED | 77.37 | 0.3510 | 0.1049 | 43.84 |
| nxEaOyG7or | 37.90 | 0.0822 | 0.0931 | 44.15 |
| OGggRA8FeM | 63.72 | 0.2918 | 0.1288 | 44.60 |
| rjEHqZ7N2X | 33.42 | 0.1282 | 0.0806 | 37.66 |
| SvA010zZI6 | 40.14 | 0.1682 | 0.1229 | 40.89 |
| u1qUWqXjWY | 44.21 | 0.0917 | 0.1050 | 22.58 |
| urVBXFAeow | 65.66 | 0.2867 | 0.1374 | 39.37 |
| VvuzvIauS6 | 44.21 | 0.1091 | 0.0982 | 45.23 |
| waXOIrgZog | 45.38 | 0.0721 | 0.1561 | 44.59 |
| BnhV1weeRf | 19.20 | 0.0246 | 0.0152 | 4.36 |
| YTQ7SppYwF | 41.67 | 0.1576 | 0.1064 | 28.42 |

