# OpenReview forum: "Improved model-based deep learning for quantitative susceptibility mapping"
_MIDL.io/2021/Conference — MIDL 2021_

### Official Review · AnonReviewer3 · 2021-03-02

**Confidence:** 3
**Preliminary Rating:** 3
**Recommendation:** Poster

**Summary:**

In this work, the authors have proposed an unsupervised learning strategy called uQSM+ for the dipole deconvolution task for the reconstruction of QSM.
This work brings in two additional terms in the loss function -  one enforcing similarity in the susceptibility values within the region of induced susceptibility and the other enforcing minimization of artifacts outside the regions of induced susceptibility values.  The authors claim that random perturbations, together with the regularized loss function improve the overall reconstruction performance.
The authors employ a prioritized training strategy by giving emphasis on dipole inversion at the early stages of training followed by gradual increase and decrease of the weights of the additional ROI loss terms to cater the regularization needs of the learning process.



**Strengths:**

The proposed approach is based on unsupervised learning and does not depend on the COSMOS labels for supervision.
The approach aims to improve the performance without increasing the computational components in the architecture
The novelty lies in the way data is augmented and fed to the network.
Extensive experimentation of the proposed method has been done both on realistic data and challenge dataset.


**Weaknesses:**

While the proposed approach could bring over a 1 dB of improvement in the PSNR compared to uQSM, SSIM has not improved at all. (Table 1)
In the case of MEDI, the PSNR improvement seems to be very high and SSIM has shown a decrease. The authors claim that MEDI maps showed substantial blurring due to heavy use of spatial regularization. In terms of structures, the visual quality appears to be similar to the baseline approaches.

To understand the actual improvements in visual quality, the authors must provide residual images with respect to the COSMOS reference for all the approaches.

This work “faintly” resembles the data augmentation approach of the MIDL 2020 paper, “Addressing The False Negative Problem of Deep Learning MRI Reconstruction Models by Adversarial Attacks and Robust Training”. (FNAF) by Cheng et al. Related work must have references based on data augmentation similar to the FNAF paper.
While the FNAP paper considers small perceptible features around the center of the image for augmenting the data, this work considers field perturbations at the periphery of the field also.

Probably the authors could analyze the metrics in a cropped region where artifacts are found and in the regions where artifacts are not found. This would help for further analysis.

The Experiment Results section is not written well.

The authors have mentioned that the deep image prior (DIP) method is compared but it is missing in Table 1.


**Deanonymize Review:**

no

**Detailed Comments:**

It is not clear why some of the metrics are expressed in % (for instance HFEN), while the values are over 100.

Since there are two terms added into the loss functions, the effect of the two terms individually is unknown. Would that have an effect on the SSIM metric?

Unable to find white arrow in Fig. 3 to understand the shadow effects. It is also not mentioned in the caption.

It was still observed that uQSM+, zs-QSM, and COSMOS results showed residual shadows  --- zs-uQSM+ instead of zs-QSM

mROI is the regions outside of bleeding and calcifications - this should be mxROI for regions outside.

The dipole convolution is perform ---> performed


**Justification Of The Preliminary Rating:**

This work shows that apart from developing the architecture for the intended task, the work considers performance improvement by data augmentation, which plays a major role in the generalization performance. Extensive experimentation has been made.
The work also considers zero-shot scenario which is likely when the datasets are less.

**Paper Type:**

methodological development

**Questions To Address In The Rebuttal:**

For the augmented local field $f_{a}$, if the susceptibility perturbations occur at the edges of the map, then would the same mask still hold good during the learning process? Or should the mask be changed to accommodate such cases?

How are the local field values for the fake bleeding and calcifications generated?

For Snr1 and Snr2, λ1 was set as 0.005 and 0.003, respectively. --- any particular reason.

What is the overall number of training patches for zs-uQSM+?



**Special Issue:**

no

---

> ### Author Response · Authors · 2021-03-17
> **Author Response**
>
> We thank the reviewer for the fair and helpful comments and spotting many unclear elaborations. Here is our detailed response:
> (1)	HFEN first uses the Laplacian of Gaussian filter to get high frequency information of two images, then calculate the normalized root mean square error of the filtered images. So, there is possible that HFEN value greater than one hundred.
> (2)	About the proposed method not showing improvement in SSIM, there are possible reasons. First, though we used COSMOS as the reference, it is not ground truth and could contain susceptibility quantification errors. Second, the proposed method did work well on structural similarity. Other reviewers also suggested using structural similarity loss in the loss function, which I will explore in the future.
> (3)	It was still observed that uQSM+, zs-QSM, and COSMOS results showed residual shadows. There are several possible reasons: (a) the local field map contained errors from field map estimation, phase unwrapping, and background field removal steps. (b) COSMOS can have errors from multi-orientation phase registration, which can introduce the QSM reconstruction errors. (c) In uQSM+ and zs-QSM+, the residual shadows could be due to the network architecture and training. We recently found that the existing zero-padding, reflective padding, and symmetric padding could bring the spatial bias and introduce some artifacts. In addition, the model loss Eq. 2 is patch based, which could also introduce some errors. We are investigating to further optimize the proposed method.
> (4)	The local field values for the fake bleeding and calcification are calculated using the physical model - dipole kernel convolution in Eq 1.
> (5)	When the susceptibility perturbations occur at the edges of the map, we still used the same mask. This could present some issues during the learning process. First, in the patch-based training, the close-to-brain-boundary region can be treated as zero-padded in the network. Recent study (MIND THE PAD – CNNS CAN DEVELOP BLIND SPOTS) found padding mechanism could bring the spatial bias. Our recent work proposed new padding for QSM, which showed better performance than existing zero-padding, reflective padding, and symmetric padding. Second, we used Eq 2 for data consistency loss. The calculated field map of estimated field map and the input field map patch is not fully consistent due to the patch-based training, which can contain inconsistency close to region boundaries. I really appreciate the reviewer pointed out this. It is necessary to investigate this part. I will do more experiment on this and add this part in the revision.
> (6)	For Snr1 and Snr2, λ1 was set as 0.005 and 0.003, respectively. The Sim1SNR1 and Sim2SNR1 datasets have higher noise level, therefore λ1 was set higher to suppress the noise in the reconstructed QSM. Since Sim1SNR2 and Sim2SNR2 datasets have lower noise level, λ1 was set lower.
> (7)	For zs-uQSM+, we only used single dataset. We used untrained network in zs-uQSM+. The network was randomly initialized. Randomly selected patches (with 50% region within brain) from the single datasets were inputted to the network, and the loss function Eq. 6 was used to update the network. After hundreds of iterations, the network converged. The network can be further fine-tuned using the full local field as input. The full local field and brain mask were inputted to get the QSM outputs.
> (8)   We will add DIP, FINE in the multi-orientation datasets evaluation.
> (8)	We will revise the figure caption,  correct the gramma errors, and improve discussion part as suggested by the reviewer.

---

### Official Review · AnonReviewer2 · 2021-03-05

**Confidence:** 3
**Preliminary Rating:** 2
**Final Rating:** 2

**Summary:**

The paper proposes an improved network architecture, uQSM+, for quantitative susceptibility mapping (QSM). It improves over its predecessor uQSM by adding data augmentation with high and low susceptibilities to the training of the network. The method was evaluated on two datasets. Results show improvements over the baseline uQSM in the first dataset. In the second dataset, a challenge, the method clearly overestimates calcifications and is outperformed by two other DL-based methods.

**Strengths:**

Overall, this paper proposes an iterative improvement over its predecessor method uQSM by adding data augmentation. The evaluation is thorough with two datasets and several baselines (although there are some inconsistencies, see weaknesses).

-	Good introduction to QSM, as well as DL-based QSM methods
-	Unsupervised method (this could be highlighted a bit more, I think)
-	Reproducible as it is intended to release the code
-	Experiments performed on two datasets, one of them public (challenge)
-	Three baselines methods (TKD, MEDI, uQSM) for the first dataset
-	Quantitative results with several metrics reported
-	Good qualitative figures to see the differences between methods

**Weaknesses:**

Most weaknesses are clarifications in the text and not of methodological nature:

-	Do I understand correctly, that uQSM+ has two forward passes, first f and m then second f_a and m, before calculating the loss? If yes, please revise the text and Fig 2 accordingly. If no, how is the network trained exactly (as in Fig 2, there are only two input channels)?
-	Experiments multi-orientation datasets: It is unclear how many scans were in total available? It is written 9 datasets and then later 40 scans from other 8 datasets. The LOOCV was performed over the 9 datasets? Please clarify! Further, the setting of the zero shot experiment is also unclear. Here one dataset is one scan? What is meant by “test data itself (…)”. Please also clarify.
-	Experiments QSM challenge: Similar to the multi-orientation, it is unclear how many cases were available for train/valid/test. Please clarify.
-	Why are the results of contrast level Sim1 omitted? Please clarify.
-	The zero shot results are quite good compared to uQSM+ (Table 1). Could you please elaborate in the discussion what the reason might be? Why are there no zero-shot results for the QSM challenge?
-	FINE seems to be state-of-the-art, what is the performance on the multi-orientation dataset?
-	Please discuss the reason for the better results of MEDI and FINE in the low susceptibility region (calcification) of the challenge case (Fig 4). What could it be? As the proposed data augmentation simulates such “spots”, might the overestimation originate from the data augmentation? Is the spot of similar size for uQSM?


**Deanonymize Review:**

no

**Detailed Comments:**

-	Text on Fig 3 (Sec 4): white arrows are mentioned but not present in Fig 3
-	Caption Fig 3: explain what the arrows show in the caption.
-	Capation Fig 4: explain what the arrows show in the caption.
-	It might be worth to show error maps between uQSM+ and COSMOS in the qualitative figures (or appendix). Differences would become more obvious.
-	Table 3 and 4: add (or highlight) the proposed method and also the ranks of the methods to the table. Also, where do the results come from (reference) and at which date were they extracted? The identifier do not help much to identify the methods, why not adding the names as in Fig 4 additionally?

The paper would benefit a proof-reading. Here are some examples:
-	Usually, commas are written after an equation when the sentence continues. E.g: Eq, where…
-	Eq 1: maybe add some space between the two equations.
-	Introduction, last paragraph: “uQSM showed promising for QSM”, the word “results” or “to be” is probably missing
-	Introduction, last paragraph: “which using” -> which uses
-	Method, 1st paragraph: “we using” -> we are using
-	Fig 1 is never referenced in the text. Also, the figure caption could explain what the reader is looking at
-	Method 2nd paragraph: “Fig 2 showed” -> shows”. Also, the Fig 2 is referenced directly twice within two sentences.
-	Eq 3 could be moved after the sentence “noise in the reconstructed susceptibility maps”
-	Sentence after Eq 5: the 2nd m_{ROI} should be m_{XROI}
-	Eq 6 could be moved after the sentence “is the weighted sum of the four losses” to improve the reading flow
-	Sec 3, 3rd paragraph: structural similarity index measure (measure missing), and abbreviation should be moved at the end of the name.
-	Sec 3, last paragraph “dataset have” -> has and “datasets” -> dataset
-	Sec 4: the full stop after “Table” is not needed
-	Discussion: “ant” -> and
-	Discussion “the how to better” -> strange wording


**Final Rating Justification:**

The authors did not respond to my review. Therefore, I cannot adjust my rating.

**Justification Of The Preliminary Rating:**

Although the paper "only" shows an iterative improvement of their method by adding data augmentation, the motivation is clear and the improvement visible (quantitative and qualitative). The authors clearly put in the effort to evaluate the proposed method on two datasets with multiple baselines. However, some sections lack clarity, which needs to be addressed for the paper to be accepted at MIDL.

**Paper Type:**

both

**Questions To Address In The Rebuttal:**

The authors should address the weaknesses that need clarifications in the text, i.e. a revision of the corresponding sections.

**Special Issue:**

no

---

> ### Author Response · Authors · 2021-03-17
> **Author Response**
>
> We thank the reviewer for the fair and helpful comments and spotting many unclear elaborations. Here is our detailed response:
> (1)	During training, uQSM+ have two forward passes to get two susceptibility maps from their respective inputs. Eq. (2) (3) (4) (5) are imposed on these two susceptibility maps to train the network.
> (2)	The multi-orientation datasets were acquired from 9 subjects. Each subject was scanned with 5 head orientations.  In uQSM+, we used leave-one-out validation, which is that for each subject, we used the other 8 subjects (8x5=40 datasets) for network training and validation. After training, the trained model was applied on the leave-out subject for testing to get QSM.
> For zs-uQSM+, we only used the normal head orientation data of each subjects. Maybe the below elaboration is clearer. We used untrained network in zs-uQSM+. The network was randomly initialized. Randomly selected patches from the single datasets were inputted to the network, and the loss function Eq. 6 was used to update the network. After hundreds of iterations, the network converged. The full local field and brain mask were inputted to get the QSM outputs.
> (3)	In stage2 of QSM challenge datasets, we used zs-uQSM+ to get susceptibility map for each dataset. We apologize that this part is not clearly written in the article. The stage2 of QSM challenge datasets have total 4 datasets, each having different image contrast and noise level. Therefore, I only implemented zs-uQSM+ in the challenging datasets.  The reason why uQSM+ is not implemented in this datasets is that the possible domain shift problem caused by different image contrast and noise level could cause the performance drop. I will do some experiments to check whether this is a big issue and add this part in the future submission.
> (4)	Why are the results of contrast level Sim1 omitted? In the official evaluation of QSM challenging, the performance was only evaluated based on signal noise level. That is the average metric scores of Sim1SN1 and Sim2SN1, Sim1SN2 and Sim2SN2, are used for performance assessment.
> (5)	We observed that MEDI and FINE performed pretty well at the calcification regions in the QSM challenging datasets, while our methods didn’t perform well. The possible reason may be that MEDI carefully tuned the regularization term to match the edge consistency between the magnitude image and the reconstructed QSM. From the submission description, FINE was trained using MEDI results as QSM target, so FINE also perform well like MEDI. In our method, we found the network has difficulties to learn extremely large susceptibility variation. The calcification region has susceptibility value -1.8ppm, while the surrounding tissue with susceptibility value about 0.02ppm. Recent studies showed networks have spectral bias and learn low frequency easier than high frequency components. Maybe this is one reason. In the future, I will explore different network architecture and loss function to investigate the problem.
> (6)	I will add FINE in the multi-orientation datasets comparison.
> (7)	I really appreciate the review pointed out the gramma errors. We will correct them!

---

### Official Review · ~Matthan_W._A._Caan1 · 2021-03-08

**Confidence:** 5
**Preliminary Rating:** 4
**Recommendation:** Oral, Poster

**Summary:**

This paper proposes data augmentation to improve QSM-quantification through Deep Learning networks. The aim is to improve reconstruction in regions with large susceptibility variations. The proposed network trained on augmented data is compared against existing methods on available challenge data. The results show the added value of embedding prior knowledge on pathological conditions in the training data/process.

**Strengths:**

Through data augmentation, this work accounts for an inherent challenge in QSM: solving an ill-posed inverse problem, which is particularly challenging in e.g. hemorrhages with a large susceptibility difference relative to its surrounding tissue. A weighted loss function is dynamically adjusted regularization during training, to emphasize consistency first outside and then inside the ROI.

**Weaknesses:**

The discussion states that FINE had a tuning loss function to quantitative metrics. Can’t a similar thing for uQSM+ be said, in that a strong spatial prior is introduced for spherically shaped bleedings? Please discuss the situation where anomalies differ in shape from the augmented objects. What bias is expected for out-of-distribution shapes? Is there experimental evidence for this in the data used in these experiments? (e.g. in vessels)

**Deanonymize Review:**

yes

**Detailed Comments:**

The paper needs a grammar revision. E.g.:
p2 ‘Though the uQSM…’ reformulate.
‘To address this issue…’ reformulate
‘we using...’
p8 ‘ahead QSM inversion’


**Justification Of The Preliminary Rating:**

The paper makes a strong case for data augmentation for robustness against anomalies in QSM. Through the use of open datasets and evaluation against established methods, the added value of the work is shown.

**Paper Type:**

methodological development

**Questions To Address In The Rebuttal:**

Experiments on other-than-spherical shapes are lacking.

**Special Issue:**

yes

---

> ### Author Response · Authors · 2021-03-17
> **Author Response**
>
> We thank the reviewer for the fair and helpful comments and spotting many unclear elaborations. Here is our detailed response:
> (1)	FINE achieved outstanding performance in the QSM challenge. From the submitted description of the method, FINE first used the MEDI result of the challenge dataset as label to train the network, and performed fine-tuning using the weighted dipole inversion loss. I checked their open code and FINE had a tuning loss function (||w(d*x-f)||^2 or ||w(exp(jd*x) - exp(jf)||^2).  I will use their code to apply on the multi-orientation datasets and have comparison.
> (2)	The reviewer gave me great comments about the possible anomalies shape bias introduced since I only used the ellipsoid shape in uQSM+. I will add this part in our future submission.
> (3)	The reviewers pointed out the other-than-spherical shapes are lacking. It could be possible that specific shape could bring bias. We are doing experiment to compare different shape - sphere, cuboid, ellipsoid, cylinder. This part will be added in the later submission.

---

### Official Review · ~Hongwei_Bran_Li1 · 2021-03-08

**Confidence:** 3
**Preliminary Rating:** 3
**Recommendation:** Oral
**Final Rating:** 2

**Summary:**

The paper is easy to follow and well written.
The authors proposed uQSM+ with data augmentation techniques to improve the model-based learning for QSM. The proposed method was evaluated on multi-orientation QSM datasets and achieved Comparable with state-of-the-art models on QSM challenge 2019 dataset.



**Strengths:**

I am not confident about the strengths, but do have a few comments:

1. A 3D-Net architecture with a data-consistency loss to regularize the training in the data augmentation schema.
2. Extensive and meaningful results.
3. Comparable with state-of-the-art models on QSM challenge 2019 dataset.


**Weaknesses:**

I am not confident about the weakness, but do have a few comments:

The main contribution is based on uQSM by the authors' previous work, the main modification is: a) batch size to meet memory constraints, and b) data augmentation in the local field to make the model robust.
Is this kind of data augmentation specially chosen? And it is not compared with other data augmentation techniques (rotation, noise... in either magnitude or phase if possible).


**Deanonymize Review:**

yes

**Detailed Comments:**

The paper is easy to follow and well written.
The main contribution is based on uQSM by the authors' previous work, the main modification is: a) batch size to meet memory constraints, and b) data augmentation in the local field to make the model robust.

A few concerns are raised when going through the paper:
1) Is this kind of data augmentation specially chosen? And it is not compared with other data augmentation techniques (rotation, noise... in either magnitude or phase if possible).
2) since it's a patch-based (96*96*96) approach, what would be the strategy to aggregate the results and get a 3D volume?
3) If I understand correctly, the training of uQSM+ has two forward passes and compares the two by a consistency loss, thus Eq. (2) (3) (4) (5) needs to be updated considering the joint optimization.

Minor and open for discussion:
1. Have the authors considered having a structural similarity loss for the magnitude image in the first loss term (Eq. 2)?
2. did the author observe any smoothing effect caused by the L1-loss for the magnitude image in Eq. 2? if yes, maybe a structural similarity loss or adversarial loss would be helpful.
3. the author mention that FINE is pre-trained on other datasets. Probably self-supervised pre-training could be helpful too.



**Final Rating Justification:**

unfortunately, the authors did not submit a rebuttal to discuss some concerns.
I could not rate accept if the discussion does not happen.

**Justification Of The Preliminary Rating:**

1. I am not mainly doing MR physics thus could not judge much on this work from MR physics.
2. A 3D-Net architecture with a data-consistency loss to regularize the training in the data augmentation schema.
3. Extensive and meaningful results.
4. Comparable with state-of-the-art models on QSM challenge 2019 dataset.


**Paper Type:**

validation/application paper

**Questions To Address In The Rebuttal:**


1. The value range of the perturbation seems not mentioned. And it's not compared with other data augmentation techniques (rotation, noise... if possible).
2. If I understand correctly, the training of uQSM+ has two forward passes and compares the two by a consistency loss, thus Eq. (2) (3) (4) (5) needs to be updated considering the joint optimization.
3. Have the authors considered having a structural similarity loss for the magnitude image in the first loss term?

**Special Issue:**

no

---

> ### Author Response · Authors · 2021-03-17
> **Author Response**
>
> We thank the reviewer for the fair and helpful comments and spotting many unclear elaborations. Here is our detailed response:
> (1)	This kind of data augmentation is specially chosen. QSM reconstruction has challenges at large susceptibility variations such as hemorrhage, calcification, and large vessels, where the low SNR causes the noise amplification at the dipole inversion (division of zeros at k-space) and causes the streaking artifacts at the reconstructed QSM. Based on our observations, uQSM also didn’t perform well at regions with large susceptibility variations. Therefore, we proposed this data augmentation method which inserting fake bleeding or calcification in the local field brought additional regularization in the unsupervised learning. The inserted bleeding has susceptibility value from 0.8ppm to 1.8 ppm, and the inserted calcification with susceptibility value from -1.8ppm to -0.8 ppm.
> (2)	Constrained by underlying physical model, traditional data augmentation techniques such as random rotation, contrast change, affine transform etc, is not applicable in this situation. In uQSM, we used image flipping at x, y directions, but is not helpful. Adding noise might work which could make the networks robust to noise. I will add this in our future submission.
> (3)	The network was trained on patch based. At inference time, the full local field and brain mask was inputted to the trained network to get the result. Though we can use the patch-based inference and stitching strategy, it could take longer processing time and might have stitching artifacts around the patch boundaries if not processed well.
> (4)	During training, uQSM+ have two forward passes to get two susceptibility maps from their respective inputs. Eq. (2) (3) (4) (5) are imposed on these two susceptibility maps to train the network.
> (5)	It is a great idea to add structure similarity loss on the magnitude images and reconstructed QSM in the performance evaluation or network training. We will explore the edge constraints (like MEDI) or MIND loss in the future work.
> (6)	In Eq. 2, we used the nonlinear dipole inversion loss for the model consistency. This is based on our uQSM work, we found the linear dipole inversion loss ||d*x-f||^2, or the weighted dipole inversion loss ||w(d*x-f)||^2 didn’t perform well. In Eq.2, the weighting parameter w could choose the noise weighting map obtained in the field map fitting from multi-echo phases or the magnitude image. We observed that it could have smoothing effect if using the magnitude image of late few echo.

---

### Meta-Review · Area_Chair1 · 2021-03-28

**Recommendation:** Accept (Poster)

**Metareview:**

The proposed paper proposes an incremental methodological improvement when compared to existing method from the author. However, the reviewers agree to find the proposed application of great interest and the paper clear and relatively well organised. Unfortunately none of the queries of the reviewers were answered during the rebuttal period

**Paper Type:**

both

---

### Decision · Program_Chairs · 2021-03-31

**Decision:**

Accept

**Comment:**

Congratulations your paper has been selected as a long oral.